Molecular identification based on mtDNA analysis of commercial crustaceans in the coastal Amazon: exotic species, cryptic diversity, and implications for sustainable fisheries in northern Brazil

Sousa Jefferson 1
Lutz Ítalo 1
Santana Paula 1
Martins Thais 1
Ferreira Charles 1
Santa Brígida Nicolly 1
Miranda Josy 1
da Silva Raimundo 2
Barbosa Andressa J. 1
Matos Suane 1
Mendes Carla 1
Cardoso Bruna 1
Silva Aline 1
da Silva Ingrid 1
da Costa Jorge 1
Vallinoto Marcelo 3
Sampaio Iracilda 3
Evangelista-Gomes Grazielle grazielle@ufpa.br graziellefeg@gmail.com 1
1 Laboratório de Genética Aplicada (LAGA), Instituto de Estudos Costeiros (IECOS), Universidade Federal do Pará , Bragança , Brazil
2 Instituto Chico Mendes de Conservação da Biodiversidade (ICMBio), Sede Tucuruí , Tucuruí , Brazil
3 Laboratório de Evolução (LEVO), Instituto de Estudos Costeiros (IECOS), Universidade Federal do Pará (UFPA) , Bragança , Brazil
Brygadyrenko Viktor
Electronic publication date: 2025 Oct 13
Publication date: 2025
Volume: 13
Electronic Location ID: e19586
Received 2024 Oct 10; Accepted 2025 May 20
Copyright: ©2025 Sousa et al.
Copyright year: 2025
Copyright holder: Sousa et al.
License: This is an open access article distributed under the terms of the Creative Commons Attribution License, which permits unrestricted use, distribution, reproduction and adaptation in any medium and for any purpose provided that it is properly attributed. For attribution, the original author(s), title, publication source (PeerJ) and either DOI or URL of the article must be cited.
License URL: https://creativecommons.org/licenses/by/4.0/

Keywords: Commercial name, DNA barcode, Exotic species, Shrimp

Funding: The Conselho Nacional de Desenvolvimento Científico e Tecnológico –CNPq (Projeto Universal n. 439113/2018-0) The Article Processing Charges (APC) were paid by the Programa de Apoio à Publicação Qualificada (PAPQ) of the Universidade Federal do Pará (UFPA) This study was financially supported by the Conselho Nacional de Desenvolvimento Científico e Tecnológico –CNPq (Projeto Universal n. 439113/2018-0) and the Article Processing Charges (APC) were paid by the Programa de Apoio à Publicação Qualificada (PAPQ) of the Universidade Federal do Pará (UFPA). The funders had no role in study design, data collection and analysis, decision to publish, or preparation of the manuscript.

==============================
Background

Located around the Caeté River estuary, the municipality of Bragança is one of the primary fishing hubs in the region. Several high-value crustacean species are intensively harvested in this area and are commonly sold at open-air markets. However, fishery products are often labeled with generic trade names, which hinders accurate species identification and conceals the true diversity of the exploited species.

Methods

Therefore, we conducted the molecular identification of crustacean species sold in Bragança. Samples were collected during two periods: from February to August 2017, and from September 2021 to May 2022. A total of 137 samples were analyzed, including 120 obtained from markets and 17 collected from the wild. Specimens were first identified morphologically, and two regions of the cytochrome c oxidase subunit I (COI) gene were amplified for molecular identification. Genetic analyses included haplotype determination, Basic Local Alignment Search Tool (BLAST) comparisons, phylogenetic tree construction, and species delimitation approaches.

Results

We obtained a dataset comprising 16 commercial names and 151 DNA sequences, including 38 sequences from region I (the barcode region) and 113 sequences from region II of the COI gene. A total of 15 crustacean species, belonging to seven genera and five families, were identified. Six of these species were classified as exotic, and three were recently described in the scientific literature. Additionally, we documented the occurrence of two distinct lineages of Penaeus monodon along the Brazilian coast. Molecular species delimitation tools effectively identified all sampled taxa and revealed underestimated levels of biodiversity due to the use of generic commercial names. This issue poses a potential threat to the long-term sustainability of fishery resources and commercial fishing in northern Brazil, as it leads to biased qualitative and quantitative assessments of fishery products.

Introduction

The northern coast of Brazil extends for over 2,500 km along the states of Amapá, Pará, and Maranhão (Ekau & Knoppers, 1999). The fauna of this region is highly diverse, encompassing both marine and estuarine species, including several taxa of significant socioeconomic and ecological importance (Camargo & Isaac, 2001; Rosa Filho et al., 2018). This remarkable biodiversity is strongly influenced by the outflow of the Amazon River, in combination with favorable environmental conditions (Barthem & Fabré, 2004; Ferreira et al., 2019).

Accordingly, the municipality of Bragança, in the state of Pará, is one of the major fish landing sites in northern Brazil, both in terms of harvested stock volume, such as crabs, and the richness of high-value species like lobsters (Furtado Júnior, Tavares & Brito, 2006). The abundance of fishery resources in this region is partly attributed to Bragança’s location within the Caeté River estuary, a river–marine system that floods surrounding mangrove areas daily, creating a nutrient-rich environment exploited by numerous aquatic species (Wolff, Koch & Isacc, 2000; Braga, Santo & Giarrizzo, 2006).

Fisheries represent a major economic activity in Bragança and surrounding communities, characterized by the exploitation of various crustacean species, including shrimp (Penaeus schmitti), swimming crabs (Callinectes spp.), and mangrove crabs (Ucides cordatus) (Vieira et al., 2014). Both artisanal (small-scale) and industrial (large-scale) fishing operations are carried out in this region, yielding significant quantities of fishery products to meet both regional and international demand (Braga, Santo & Giarrizzo, 2006).

Several crustacean species have been commonly reported in local commerce (Freire, Silva & Souza, 2011; Santana et al., 2020), with commercial practices largely based on the use of trade names (popular nomenclature) and sales categories (Santana et al., 2020). As a result, distinct species are often sold under the same common name, while a single species may be marketed under multiple names (Santana et al., 2020; Santana et al., 2023), complicating the proper management of fishery stocks. In a survey of species sold at the “feira livre” (street market) in Bragança. Freire, Silva & Souza (2011) identified eight crustacean species being marketed under three generic categories: “camarão” (shrimp) (n = 5), “siris” (crabs) (n = 2), and “caranguejo” (mangrove crab) (n = 1). More recently, Santana et al. (2020) recorded 14 trade names corresponding to only seven crustacean species sold at the same location.

Popular nomenclature is an unreliable parameter for assessing regional biodiversity, as most common names typically refer to a group of species (Santana et al., 2020). This leads to biased estimates of fishery stocks and species richness, as previously demonstrated for bony fishes and elasmobranchs (Martins et al., 2021; Santana et al., 2023). Furthermore, the studies by Freire, Silva & Souza (2011) and Santana et al. (2020) relied on morphological species identification, a limited approach primarily due to the morphological similarities between taxa and the lack of comprehensive literature covering all developmental stages. Consequently, more accurate techniques are necessary to properly assess the true species diversity exploited in fisheries, such as species-specific DNA sequences from the mitochondrial genome.

Among the mitochondrial DNA (mtDNA) markers available, partial sequences of the cytochrome c oxidase subunit I (COI) gene have been widely used for species identification across various animal groups, including crustaceans (Silva-Oliveira et al., 2011). The initial COI region (∼650 base pairs or bp) has been officially designated as a global standard for animal identification, known as DNA barcoding (Hebert et al., 2003), and has been successfully applied in fisheries authentication (Pejovic et al., 2016; Bezeng & Bank, 2019; Santana et al., 2023). Furthermore, both the first and second regions of the COI gene have proven effective in distinguishing crustacean species (Silva-Oliveira et al., 2011; Ferreira et al., 2023). For example, several studies based on sequencing the second COI region have yielded reliable and unequivocal results (Silva-Oliveira et al., 2011; Udayasuriyan et al., 2015; Harris, Rosado & Xavier, 2016), particularly considering that available COI datasets for crustaceans remain limited to a few taxa. However, in certain cases, the exclusive use of mtDNA can present constraints and limitations, such as maternal inheritance and a predisposition to introgression, which may lead to biased conclusions (Mutanen et al., 2016; Kochanova, 2024).

Given the importance of fishing resources and the limited knowledge regarding the true diversity of commercial species in the Bragança region, Amazon coast, we utilized the COI gene to identify crustacean species sold under different commercial names, including samples obtained by Santana et al. (2020) and those from natural environment locations. These data will also contribute to the creation of a reference library for the carcinofauna of the Amazon coast, aiming to expand public COI databases, with an emphasis on the barcode region. In addition to generating technical and scientific knowledge, our goal is to provide a foundation for the development of public policies that ensure the revenue from commercial fishing is linked to the sustainability of fishery resources and food security on the Amazon coast.

Materials and Methods

Collection and processing of samples

Crustacean specimens from the Bragança market were obtained during two sampling periods. The first set of samples was collected between February and August 2017 as part of a study by Santana et al. (2020). Taxa were identified morphologically using species identification keys (Melo, 1996; Costa et al., 2003) and subsequently labeled according to their commercial trade names. The second sampling period extended from September 2021 to May 2022. During both periods, fresh and processed specimens (cooked or salted) were collected biweekly (every 15 days) and recorded based on their commercial labels or local names provided by traders. Combined, these efforts resulted in 16 months of sampling, covering the crustaceans marketed in Bragança (1°03′57″S, 46°47′22″W; Fig. 1) across more than a full annual cycle.

Figure 1 Location of the Bragança Free Fair (“Feira livre”) in coastal Amazon (red circle).

(A) Location of the open-air market, with the dashed line indicating the area where crustaceans are sold. (B) Common methods of selling crustaceans observed in the Bragança Free Fair.

In contrast, samples from natural environments were collected on the Ajuruteua Peninsula between November 2021 and June 2022, with biweekly sampling intervals. Crustaceans in this region were captured by local fishermen using trawl nets (25 mm mesh size). Specimens were initially documented according to the local names reported by the fishermen (Fig. 1) and later identified morphologically using the previously cited taxonomic keys (Melo, 1996; Costa et al., 2003). Overall, crustacean diversity was assessed using 137 sequenced samples, comprising 120 specimens sourced from the street market and 17 collected from wild populations (Table S1).

Tissue samples were extracted from each specimen, preserved in 2.0 mL microtubes containing 90% ethanol, and labeled with unique registration numbers. The samples were then stored at −20 °C for long-term preservation in the Laboratório de Genética Aplicada (LAGA).

All specimens were morphologically identified using taxonomic keys (Melo, 1996; Costa et al., 2003) along with their commercial trade names. The specimens were then photographed following standard scientific protocols, preserved according to the methods outlined by Martins (1994), and deposited as voucher specimens in the LAGA fisheries collection (Instituto de Estudos Costeiros, Universidade Federal do Pará, Brazil). These contributions enhance the regional catalog of commercial carcinofauna.

Laboratorial procedures

Genomic DNA was extracted from each tissue sample using the Wizard Genomic DNA Purification Kit (Promega) or the NaCl protocol adapted from Aljanabi & Martinez (1997). After isolation, the DNA was stained with a solution containing GelRed™ (Biotium) and blue juice buffer (three µL of solution: three µL of DNA) and subjected to electrophoresis on a 1% agarose gel at 60 V for 30 min. After this period, the DNA samples were visualized under UV light to assess the quality of the isolated DNA. Finally, DNA quantification was performed using a NanoDrop® 2000 spectrophotometer.

The COI sequences were amplified via PCR (polymerase chain reaction) using 2.4 µL of dNTPs (1.25 mM), 1.5 µL of buffer (10x), 0.6 µL of MgCl2 (50 mM), 0.6 µL of each primer (50 ng/µL), approximately 100 ng of template DNA, 0.1 µL of Taq DNA polymerase (5 U/µL), and ultrapure water to a final volume of 15 µL.

The molecular identification of samples proved particularly challenging due to difficulties in standardizing the PCR reactions (reaction stoichiometry and cycling conditions) for the barcode region of many specimens. Additionally, during the development of this study, only a few public barcode sequences were available for commercial crustaceans. Therefore, two COI regions were used in this study. The first is the traditional barcode region, referred to in this study as “region I”, amplified using primers LCO1490 and HCO2198 (Folmer et al., 1994), and the second is the second COI region, referred to as “region II”, amplified using primers COIA and COIF (Palumbi & Benzie, 1991). Both COI regions were amplified under the same conditions, as follows: an initial denaturation step at 94 °C for 4 min, followed by 35 cycles of denaturation at 95 °C for 35 s, annealing at 48°–59 °C for 40 s, and extension at 72 °C for 45 s, with a final extension step at 72 °C for 5 min.

The amplicons were purified using PEG 8000 (polyethylene glycol) according to the method described by Paithankar & Prasad (1991), and then subjected to the sequencing reaction using the dideoxy-terminal method (Sanger, Nicklen & Coulson, 1977) with the Big Dye kit (ABI PrismTM Dye Terminator Cycle Sequencing Reaction; Thermo Fisher Scientific, Waltham, MA, USA). After precipitation, the products were analyzed on an automatic sequencer (ABI 3500; Thermo Fisher Scientific).

Sequence datasets and genetic analyses

The DNA sequences were aligned and edited using BioEdit software (Hall, 1999), where potential errors or uncertain nucleotides were identified and corrected. Subsequently, the sequences were automatically aligned using the Clustal-W package (Thompson, Higgins & Gibson, 1994), also integrated within BioEdit (Hall, 1999). A second round of visual correction was performed on the final alignments, when necessary. Based on this final dataset, haplotypes were determined using DNAsp software (Librado & Rozas, 2009) to aid in the identification of the samples. The sequences generated in this study have been deposited in NCBI. Region I is available under accession numbers PP706442 to PP706467, while region II is available under accession numbers PP708622 to PP708656.

To determine the levels of genetic similarity among the samples for species identification, each haplotype was compared to public sequence datasets available on the GenBank platform using the Basic Local Alignment Search Tool (BLAST) for nucleotides (Altschul et al., 1997), as well as on the BOLD (Barcoding of Life Database) system (Ratnasingham & Hebert, 2007). The published sequences showing the highest similarity to the samples in this study were included in the sequence datasets used to generate the cladograms. Additionally, we assessed the number of polymorphic sites, which may indicate species-specific mutations, and examined the presence of putative stop codons using the tools available in MEGA 11.0.11 (Tamura, Stecher & Kumar, 2021).

Phylogenetic analyses were performed using two separate datasets, each corresponding to a distinct region of the COI gene. This approach was selected because the concatenated dataset did not produce reliable phylogenetic topologies, likely due to the limited number of sequences and haplotypes in region I, which caused a discrepancy in sequence numbers between the two regions.

To corroborate the levels of similarity and infer the phylogenetic relationships among the analyzed samples, phylogenetic trees were constructed using Bayesian inference (BI) and maximum likelihood (ML) approaches.

The BI analysis was performed using BEAST 1.10.4 software (Drummond et al., 2012). The parameters for the BI included the use of a strict clock and Yule speciation prior. Posterior probabilities were estimated assuming 20 million generations with a 10% burn-in. The analytical parameters were inspected in Tracer 1.5 (Rambaut & Drummond, 2009) to evaluate the convergence of chains, with Effective Sampling Size (ESS) values above 200 considered adequate. The trees generated in BEAST were summarized using TreeAnnotator 1.10.4 software (Suchard et al., 2018) to obtain a consensus tree. The ML analysis was conducted using IQ-TREE 2.1.3 software (Nguyen et al., 2015) and was based on 1,000 pseudoreplicates to ensure statistical reliability (Felsenstein, 1985). In both analyses (BI and ML), the evolutionary model GTR+I was used, as determined by JModelTest 2 (Darriba et al., 2012).

In MEGA 11.0.11 (Tamura, Stecher & Kumar, 2021), we generated pairwise genetic distance matrices for both intra and interspecific comparisons based on the K2P model (Kimura, 1980), which is the standard molecular evolution model commonly used in DNA barcoding studies (Hebert et al., 2003)

Finally, the software FigTree 1.4.4 was used to visualize the final trees (Rambaut, 2014), followed by editing in InkScape 0.92.4 (https://www.inkscape.org). The diagram illustrating the relationships between common trade names and the identified species was created using RAWGraphs (https://rawgraphs.io).

Species delimitation

Species delimitation tests were conducted separately for each genetic region (I and II) using three complementary methods: the Generalized Mixed Yule Coalescent (GMYC; Pons et al., 2006), the Bayesian Poisson Tree Process (bPTP; Zhang et al., 2013), and Assemble Species by Automatic Partitioning (ASAP; Puillandre, Brouillet & Achaz, 2021). These analyses aimed to define molecular operational taxonomic units (MOTUs), which represent potential species-level entities (Floyd et al., 2002).

The GMYC analysis was conducted using the Splits package (Ezard, Fujisawa & Barraclough, 2009) within the R 4.4.2 environment (R Core Team, 2024). An ultrametric tree, generated in BEAST 1.10.4 (Drummond et al., 2012), was used for this analysis.

The bPTP method was performed using a ML tree as the input file, generated in IQ-TREE 2.1.3 software (Nguyen et al., 2015). The bPTP analysis was executed on the online PTP platform (https://species.h-its.org/ptp/) using the platform’s default parameters.

The ASAP analysis was conducted using the online ASAP platform (https://bioinfo.mnhn.fr/abi/public/asap/asapweb.html), based on a K2P distance matrix generated in MEGA 11.0.11 (Tamura, Stecher & Kumar, 2021), using the Kimura (K80) TS/TV evolutionary model.

Results

Characterization of datasets and trade names

The sampling efforts yielded a total of 250 specimens, representing the primary categories of crustaceans: “camarão” (shrimp), “lagosta” (lobster), “siri” (crabs), and “caranguejo” (mangrove crab). A total of 16 trade names were recorded, with the majority corresponding to shrimp, represented by 12 popular trade names, including: “camarão branco”, “camarão cascudo”, “camarão piré”, “camarão vananeio”, “camarão grazado”, “camarão piticaia”, “camarão vermelho”, “camarão bate pé”, “camarão tigre”, “camarão arco-íris”, “camarão da Amazônia” and “camarão pitu”. In the case of crabs, no subdivisions within the general categories “caranguejo” and “siri” were recorded. As for lobsters, two distinct trade names were reported: “lagosta vermelha” and “lagosta sapata”.

Molecular identification based on genetic similarity levels

The final dataset for region I (barcode region) included 38 sequences of 560 bp, corresponding to 26 haplotypes (Table 1), representing the following species: P. schmitti, Penaeus vannamei, Penaeus isabelae, Penaeus monodon lineage 1 (clade A), Penaeus monodon lineage 2 (clade B), Xiphopenaeus dincao, Xiphopenaeus kroyeri, Mierspenaeopsis sculptilis, Macrobrachium equidens, U. cordatus, Callinectes bocourti, Panulirus meripurpuratus, and Panulirus laevicauda. On the other hand, the dataset for region II consisted of 113 sequences of 425 bp, representing 35 haplotypes. These haplotypes corresponded to the following species: P. schmitti, P. vannamei, P. isabelae, P. monodon L1, P. monodon L2, X. dincao, X. kroyeri, Macrobrachium rosenbergii, M. sculptilis, M. equidens, U. cordatus, C. bocourti, and Callinectes danae.

Table 1 The frequency of each haplotype is shown in parentheses.

The asterisks refer to the samples with successful sequencing of both COI regions.

Haplotype	COI region	Reference sample	Location	Commercial designation	NCBI/BOLD identity and access code	NCBI/BOLD similarity	
1 (1)	I	cbr01	“Feira Livre”	“Camarão branco”	Penaeus schmittiMT607578/PENAE056	100%/100%	
2 (1)	I	cca14	“Feira Livre”	“Camarão cascudo”	Penaeus vannameiMW027142/GBMNB54360	100%/100%	
3 (1)	I	cbr55	“Feira Livre”	“Camarão branco”	Penaeus isabelaeMG662009/GBCMD31194	99.82%/99.82%	
4 (1)	I	cca10	“Feira Livre”	“Camarão cascudo”	Penaeus isabelaeMG662009/GBCMD31194	98.93%/98.92%	
5 (2)	I	ct04	“Ajuruteua	“Camarão tigre”	Penaeus monodon clade A MT449918/GBMNF10111	100%/100%	
6 (1)	I	ct05*	“Ajuruteua	“Camarão tigre”	Penaeus monodon clade B MT449922/GBMNC67825	100%/100%	
7 (1)	I	cbr19	“Feira Livre”	“Camarão branco”	Xiphopenaeus dincaoKY449120/GBCMD25426	100%/100%	
8 (3)	I	cpi04	“Feira Livre”	“Camarão branco”	Xiphopenaeus kroyeriKY449079/GBCMD25385	100%/100%	
9 (3)	I	cpi13	“Feira Livre”	“Camarão piré”	Xiphopenaeus kroyeriKY449079/GBCMD25385	99.64%/99.64%	
10 (2)	I	carc05*	“Feira Livre”	“Camarão arco-íris”	Mierspenaeopsis sculptilisKP297897/GBMNC67910	100%/100%	
11 (3)	I	cam01*	“Feira Livre”	“Camarão da Amazônia”	Macrobrachium rosenbergiiOL824984/ANGEN102	100%/ 100%	
12 (2)	I	cca01*	“Feira Livre”	“Camarão cascudo”	Macrobrachium equidensMW479976/GBMND16080	99.82%/99.82%	
13 (3)	I	car01	“Feira Livre”	“Caranguejo”	Ucides cordatusKU313508/GBCMD22808	99.82%/100%	
14 (2)	I	car02	“Feira Livre”	“Caranguejo”	Ucides cordatusKU313508/GBCMD22808	100%/99.82%	
15 (1)	I	car12*	“Feira Livre”	“Caranguejo”	Ucides cordatusKU313508/GBCMD22808	99.29%/99.10%	
16 (1)	I	car15*	“Feira Livre”	“Caranguejo”	Ucides cordatusKU313508/GBCMD22808	99.46%/99.28%	
17 (1)	I	car16*	“Feira Livre”	“Caranguejo”	Ucides cordatusKU313508/GBCMD22808	99.64%/99.46%	
18 (1)	I	sir05	“Feira Livre”	“Siri”	Callinectes bocourtiMG462542/GBCMD28602	99.11%/99.10%	
19 (1)	I	sir06	“Feira Livre”	“Siri”	Callinectes bocourtiMG462542/GBCMD28602	99.64%/99.64%	
20 (1)	I	sir08	“Feira Livre”	“Siri”	Callinectes bocourtiMG462542/GBCMD28602	100%/100%	
21 (1)	I	sir09	“Feira Livre”	“Siri”	Callinectes bocourtiMG462542/GBCMD28602	99.82%/99.82%	
22 (1)	I	sir01	“Feira Livre”	“Siri”	Callinectes bocourtiMG462542/GBCMD28602	99.82%/99.82%	
23 (1)	I	sir02*	“Feira Livre”	“Siri”	Callinectes bocourtiMG462542/GBCMD28602	99.82%/99.82%	
24 (1)	I	lag01	“Feira Livre”	“Lagosta vermelha”	Panulirus meripurpuratusMF490043/GBCMD28243	99.46%/99.46%	
25 (1)	I	lag02	“Feira Livre”	“Lagosta vermelha”	Panulirus meripurpuratusMF490043/GBCMD28243	99.46%/99.46%	
26 (1)	I	lsa03	“Feira Livre”	“Lagosta sapata”	Panulirus laevicaudaAF339462/Early-Release	98.21%/99.10%	
1 (10)	II	cbr18	“Feira Livre”	“Camarão branco”	Penaeus schmittiAY135189	100%	
2 (2)	II	cbr17	“Feira Livre”	“Camarão branco”	Penaeus schmittiAY135189	99.76%	
3 (1)	II	cbr05	“Feira Livre”	“Camarão branco”	Penaeus schmittiAY135189	99.53%	
4 (7)	II	cgr03	“Feira Livre”	“Camarão grazado”	Penaeus vannameiMT178583	100%	
5 (3)	II	cve02	“Feira Livre”	“Camarão vermelho”	Penaeus vannameiMT178583	99.76%	
6 (2)	II	cca23	“Feira Livre”	“Camarão cascudo”	Penaeus vannameiMT178583	99.53%	
7 (11)	II	cba02	“Feira Livre”	“Camarão bate pé”	Penaeus isabelaeMN240522	100%	
8 (19)	II	cpi10	“Feira Livre”	“Camarão piré”	Penaeus isabelaeMN240522	99.29%	
9 (1)	II	cca33	“Feira Livre”	“Camarão cascudo”	Penaeus isabelaeMN240522	99.76%	
10 (1)	II	cca07	“Feira Livre”	“Camarão cascudo”	Penaeus isabelaeMN240522	99.53%	
11 (1)	II	cbr42	“Feira Livre”	“Camarão branco”	Penaeus isabelaeMN240522	98.82%	
12 (1)	II	cts04	“Feira Livre”	“Camarão tigre”	Penaeus monodon clade A KX459267	99.76%	
13 (1)	II	ct05	“Ajuruteua”	“Camarão tigre”	No correspondence	No data	
14 (7)	II	cgr02	“Feira Livre”	“Camarão grazado”	Xiphopenaeus kroyeriDQ084368	100%	
15 (7)	II	cpi44	“Ajuruteua”	“Camarão piré”	Xiphopenaeus kroyeriDQ084368	99.76%	
16 (1)	II	cpi46	“Ajuruteua”	“Camarão piré”	Xiphopenaeus kroyeriDQ084368	99.53%	
17 (1)	II	cpi12	“Feira Livre”	“Camarão piré”	Xiphopenaeus kroyeriDQ084368	99.76%	
18 (4)	II	cpi39	“Ajuruteua”	“Camarão piré”	Xiphopenaeus dincaoDQ084376	100%	
19 (2)	II	cpic01	“Feira Livre”	“Camarão piticaia”	Xiphopenaeus dincaoDQ084376	99.82%	
20 (9)	II	carc01	“Feira Livre”	“Camarão arco-íris”	Mierspenaeopsis sculptilisMT178686	97.65%	
21 (2)	II	cam02	“Feira Livre”	“Camarão da Amazônia”	Macrobrachium equidensKM255682	100%	
22 (1)	II	cpi43	“Feira Livre”	“Camarão piré”	Macrobrachium equidensKM255682	99.53%	
23 (1)	II	cca02	“Feira Livre”	“Camarão cascudo”	Macrobrachium equidensKM255682	99.53%	
24 (3)	II	cam01	“Feira Livre”	“Camarão da Amazônia”	Macrobrachium rosenbergiiMK782972	100%	
25 (1)	II	cpit01	“Feira Livre”	“camarão pitu”	Macrobrachium rosenbergiiMK782972	99.53%	
26 (2)	II	car13	“Feira Livre”	“Caranguejo”	No correspondence	No data	
27 (1)	II	car11	“Feira Livre”	“Caranguejo”	No correspondence	No data	
28 (1)	II	car12	“Feira Livre”	“Caranguejo”	No correspondence	No data	
29 (1)	II	car15	“Feira Livre”	“Caranguejo”	No correspondence	No data	
30 (2)	II	car16	“Feira Livre”	“Caranguejo”	No correspondence	No data	
31 (1)	II	car06	“Feira Livre”	“Caranguejo”	No correspondence	No data	
32 (1)	II	car07	“Feira Livre”	’Caranguejo	No correspondence	No data	
33 (3)	II	sir11	“Ajuruteua”	“Siri”	Callinectes danaeMH062450	99.29%	
34 (1)	II	sir14	“Ajuruteua”	“Siri”	Callinectes danaeMH062450	98.82%	
35 (1)	II	sir02	“Feira Livre”	“Siri”	No correspondence	No data	
Notes.

* Samples with successful sequencing of both COI regions.

Most species were represented in both COI datasets, with the exception of the lobsters from the genus Panulirus (found only in region I) and the crab C. danae (found only in region II). All crustacean taxa were successfully discriminated, resulting in a total of 15 species from seven genera and five families, including a possible new species of Penaeus monodon. The haplotype similarity levels from both datasets in relation to the public datasets ranged from 98% to 100% (Table 1). The family Penaeidae contained the highest number of species (n = 7), followed by Palaemonidae, Portunidae, and Palinuridae, each with two species. Within Penaeidae, Penaeus was the most diverse genus, represented by four species, while one or two species were reported for the other genera.

Some taxa were differentiated by region II; however, due to the absence of reference sequences in NCBI for these taxa, they were identified using region I (the barcode region). These taxa include U. cordatus, C. bocourti, and lineage 2 of P. monodon. The species identified from the wild-caught samples included P. schmitti, P. isabelae, P. monodon clade A, P. monodon clade B, X. kroyeri, X. dincao, and C. danae. Six of the 15 recorded species were alien species (P. vannamei, P. monodon clade A, P. monodon clade B, M. sculptilis, M. equidens, and M. rosenbergii), while three represented recently described taxa (P. isabelae, X. dincao, and P. meripurpuratus).

Identification based on phylogenetic reconstruction

The BI and ML trees yielded similar topologies, both recovering the same reciprocally monophyletic groups and clustering the haplotypes of each species with high support values. Therefore, the BI tree was selected to represent the phylogenetic inferences, while the ML trees are presented in Fig. S1. In the tree corresponding to region II, which included the 35 haplotypes, all taxa were discriminated. However, only 10 species were identified, as no public reference sequences were available for some of the samples (Fig. 2).

Figure 2 Bayesian inference (BI) tree showing the haplotypes of crustaceans based on the second region (region II) of the Cytochrome C Oxidase Subunit I (COI) gene, including the reference sequences from public databases.

The statistical support values are shown on internodes.

In the tree corresponding to region I, the 26 haplotypes formed 14 clades, each representing a species, with public reference sequences (Fig. 3). This analysis included the three taxa that were not previously identified by COI region II, as follows: “caranguejo” (U. cordatus), “siri” (C. bocourti), and one lineage of “camarão tigre” (P. monodon L2). The two lineages identified as “camarão tigre” (referred to as P. monodon clades A and B) were differentiated by genetic distance values above 7.23%, thus resulting in a total set of 15 species.

Figure 3 Bayesian inference (BI) tree showing the haplotypes of crustaceans based on the first region (barcode region or region I) of the Cytochrome C Oxidase Subunit I (COI) gene, including the reference sequences from public databases.

The statistical support values are shown on internodes.

The highest intraspecific distances observed for region I were found in P. isabelae (1.27%), U. cordatus (1.27%), and C. bocourti (1.27%) (Table S2). For region II, the highest values were recorded in P. isabelae (1.43%), U. cordatus (1.67%), and X. dincao (1.19%) (Table S3). Some samples showed significant intraspecific distance when compared to sequences from the public database, such as the “lagosta” sample (lsa03) diverging by 1.82% from P. laevicauda (AF339462) in region I (Table S2), and the Carc05 sample showing a 2.40% divergence from the M. sculptilis sequence (MT178686) in region II (Table S3).

The interspecific distance values ranged from 7.23% to 34.01%, with the smallest distance observed between P. monodon clades A and B for region I (Table S2). For region II, the distances ranged from 10.7% to 32.56%, with the smallest value also observed between P. monodon clades A and B (Table S3).

The delimitation tests for region I identified 14 MOTUs across all three methods (ASAP, bPTP, and GMYC). For region II, ASAP and GMYC delimited 13 MOTUs, while bPTP identified 14 MOTUs, including the separation of Carc05 from the M. sculptilis sequence (MT178686). All tests consistently identified P. monodon (clade A and clade B) as distinct MOTUs.

Commercial trade names versus species

As mentioned earlier, “camarão” (shrimp) was the commercial category with the highest number of trade names, encompassing 12 popular designations (Fig. 4). For the wild-collected crustacean samples, the common names included “camarão branco,” “camarão piré,” “camarão tigre,” and “siri”.

Figure 4 Alluvial diagram representing the 15 species of crustaceans recorded on the coastal Amazon.

Relationships among popular trade names (left), scientific species names (center), and respective families (right). Matching colors illustrate connections among common names, species, and families. The thickness of the connecting lines represents the frequency with which each popular name is used.

Except for P. schmitti, all other species of Penaeidae and two species of Macrobrachium were commercialized under different trade names. The species with the most recorded designations were P. vannamei (“camarão cascudo”, “vananeio”, “grazado”, “piticaia”, and “vermelho”) and P. isabelae (“camarão branco”, “cascudo”, “piré”, and “bate pé”) (Fig. 4). On the other hand, the term “siri” (crabs) and six shrimp trade names were used to refer to more than one species. For instance, “camarão branco” and “camarão piré” were trade names used to refer to four distinct species each (Fig. 4).

Representative images of the crustacean species (P. schmitti, P. vannamei, P. monodon, X. kroyeri, M. sculptilis, M. rosenbergii, P. meripurpuratus , C. bocourti, and U. cordatus) identified within each category of trade names are shown in Fig. 5.

Figure 5 Representative species from each category of crustaceans identified by morphological traits, commercial names and DNA marker (Cytochrome C Oxidase Subunit I).

Discussion

Diversity of commercialized crustaceans validated by the COI marker

The COI marker proved to be an effective tool in uncovering the diversity of the commercial carcinofauna, representing the first dataset of mitochondrial DNA sequences for identifying commercialized crustaceans in the coastal Amazon. This DNA barcoding approach for assessing crustacean diversity has also been applied in various geographic regions, including the Gulf of Mexico (Varela et al., 2021), Southern Africa (Bezeng & Bank, 2019), the North Sea (Raupach et al., 2015), Southeast Asia (Hurzaid et al., 2020), Malaysia (Jamaluddin et al., 2019), and Taiwan (Huang & Shih, 2021). In all these regions, the molecular tool has proven effective and accurate for species identification. Through comparative analyses with public databases and phylogenetic tree topologies, we identified a total of 15 species, including cryptic diversity within P. monodon, which has been previously reported in the literature (Yudhistira & Arisuryanti, 2019).

The number of crustacean species recorded in the present study exceeds those reported by other authors in the same region (Freire, Silva & Souza, 2011; Santana et al., 2020). This discrepancy is likely due to our use of molecular species delimitation tools, whereas previous studies relied on external morphological traits for species identification. This result underscores the limitations of morphological characteristics in accurately estimating species richness. Notably, we analyzed the same samples collected by Santana et al. (2020), but identified a higher number of valid species, including the potential new species of the exotic “camarão tigre.” We, therefore, recommend that future studies in integrative taxonomy incorporate DNA-based methods to establish a reliable inventory of commercial species.

It is important to note that the COI region II sequences of certain taxa did not exhibit species-level similarities when compared to the NCBI public database, as public sequences for this region are available only for a limited number of crustacean species (Weigand et al., 2019; Dwiyitno, Hoffman & Parmentier, 2022). Nevertheless, the number of species identified using region II further supports the potential of the entire COI gene (regions I and II) as an efficient DNA marker for species delimitation. However, region I remains the most widely used and accepted fragment for this purpose (Silva-Oliveira et al., 2011; Veneza et al., 2014; Collin et al., 2020; França et al., 2021).

We observed the presence of 10 shrimp species (families Penaeidae and Palaemonidae), two species of “siri” crabs (genus Callinectes), two species of lobsters (genus Panulirus), and one species of mangrove crab (U. cordatus) being sold at the Bragança street market. In contrast, Santana et al. (2020) recorded five shrimp species, along with a single species of “siri” crab (C. bocourti) and mangrove crab (U. cordatus). Freire, Silva & Souza (2011) identified five shrimp species, two “siri” crab taxa (C. danae and C. bocourti), and one mangrove crab species (U. cordatus). Notably, the shrimp species P. isabelae, P. monodon (clades A and B), X. dincao, M. sculptilis, and the two lobster species (genus Panulirus) were not recorded in previous studies. Regarding the species documented at the Bragança fishing landing by Espírito-Santo & Isaac (2012), the only species absent from their list were the exotic and newly described ones.

On the other hand, Freire, Silva & Souza (2011) identified native species of freshwater prawns from the genus Macrobrachium, such as M. surinamicum and M. amazonicum, being commercialized in Bragança. These species were not recorded in the present study, although M. amazonicum is considered the most exploited species of native freshwater prawn in the region (Bentes et al., 2011). It is likely that specimens of M. amazonicum were present among the non-sequenced samples, as the trade name “camarão da Amazônia” (Amazon freshwater prawn) was recorded at the collection sites. Instead, all Macrobrachium sequences obtained from the samples commercialized in Bragança referred to exotic species.

Regarding crabs “siri” (genus Callinectes), two species are found in the Bragança region: C. danae and C. bocourti (Bentes et al., 2013). Both species were sampled in the present study, but only C. bocourti was found in the open market. In fact, C. danae was not recorded by Santana et al. (2020), as local vendors reported a decrease in commercial demand for this species. The last recorded presence of C. danae in the local market was in the earlier study by Freire, Silva & Souza (2011).

Genetic distances and delimitation tests

The three species delimitation tests identified two molecular operational taxonomic units (MOTUs) within P. monodon, which aligns with the high genetic distances observed in both region I (7.23%) and region II (10.7%). Haplotypes CT4 and CT5 in region I show 100% similarity with sequences MT449918 (Clade A) and MT449922 (Clade B), respectively (Table 1 and Fig. 2). This suggests that these haplotypes belong to the same genetic stock and represent the same cryptic species previously reported by Yudhistira & Arisuryanti (2019). Other studies have also highlighted cryptic diversity in P. monodon (Chan, Muchlisin & Hurzaid, 2021; Ramirez et al., 2021), indicating the possible existence of up to four new species yet to be described (Hurzaid et al., 2020; Farias et al., 2023).

For M. sculptilis, the species delimitation tests produced varying results depending on the region analyzed. In region I, the tests were consistent, identifying a single MOTU for the Carc05 haplotype and the KP297897 sequence, with a genetic distance of 0%. However, in region II, the GMYC and ASAP methods identified one MOTU, while bPTP identified two MOTUs, reflecting the intraspecific genetic distance of 2.40% observed between the Carc05 haplotype and the M. sculptilis sequence (MT178686). This suggests the presence of distinct lineages, consistent with previous reports of cryptic species within M. sculptilis (Hurzaid et al., 2020). Notably, the sequences from public databases used in the analyses of each region originate from different geographic locations, which likely contributed to the differing outcomes of the tests.

In the literature, P. isabelae is considered a cryptic species, with delimitation tests revealing the existence of two MOTUs in Brazil (Ramirez et al., 2021; Farias et al., 2023), with a minimum genetic distance of 1% between them (Ramirez et al., 2021). However, in our study, individuals of P. isabelae, exhibiting genetic divergence ranging from 1.27% to 1.43% (regions I and II), were recovered as a single MOTU across all delimitation tests.

The species U. cordatus exhibited intraspecific genetic distances of 1.27% in region I and 1.67% in region II, and was identified as a single MOTU. The literature has documented U. cordatus as having high genetic diversity, substantial gene flow, and a single panmictic population (Buranelli, Felder & Mantelatto, 2019).

Identification and occurrence of new species of crustaceans in the regional market

Most of the shrimps traded in Bragança consist of previously recorded alien species (Maciel et al., 2011; Cintra et al., 2015; Ferreira et al., 2023). Among these, the “camarão arco-íris” or rainbow shrimp (M. sculptilis) has been recently reported by Ferreira et al. (2023) based on DNA barcode identification of eight specimens. This species is native to the Indo-Pacific region, and the authors hypothesized that its invasion originated from ballast water and biofouling on large ships (Ferreira et al., 2023).

Another alien species found in the region is P. vannamei, a native shrimp from the Eastern Pacific that was introduced to Brazil for aquaculture in the 1970s (Loebmann, Mai & Lee, 2010). Although not recorded in ecosystems along the coastal Amazon, escapes of captive specimens into the wild have been reported in the Parnaíba Delta, northeastern Brazil (Loebmann, Mai & Lee, 2010). Similarly, M. rosenbergii, originally from southeastern Asia, was introduced in 1977 to support shrimp farming in Brazil (Cavalcanti, 1998; Oliveira & Santos, 2021), and is now present in several Brazilian river basins due to accidental escapes (Iketani et al., 2016; Oliveira & Santos, 2021). Another alien species of Macrobrachium, M. equidens, native to the Indo-Pacific Ocean, was first recorded along the coastal Amazon by Maciel et al. (2011), though there is no available information on the timing or method of its introduction (Gomes et al., 2014).

Similarly, the tiger prawn P. monodon, an Indo-Pacific species, was introduced to Brazil in the 1970s for shrimp farming (Leão et al., 2011). However, many specimens escaped, leading to the establishment of several populations in estuarine and coastal areas of northern and northeastern Brazil (Coelho, Santos & Ramos-Porto, 2001; Cintra et al., 2015). In the present study, we identified two genetically divergent lineages of P. monodon (>7%) co-occurring in sympatry. This represents the first record of such a phenomenon in the coastal Amazon, complementing earlier findings documented in Indonesia (Yudhistira & Arisuryanti, 2019).

Furthermore, we recorded three recently described species. These include X. dincao (Carvalho-Batista et al., 2019), which was previously referred to as Xiphopenaeus spp. II by Gusmão et al. (2006), and P. isabelae (Tavares & Gusmão, 2016), which was once regarded as morphotype I of P. subtilis (Gusmão, Lazoski & Solé-Cava, 2000). Since we sampled several specimens of P. isabelae and none of P. subtilis, we infer that in previous reports from this region, the identification of P. subtilis based on morphology likely referred to P. isabelae, as both species exhibit high morphological similarities and low levels of genetic divergence (França et al., 2021).

The third recently described species was recorded among the lobster samples (“lagosta”), identified as P. meripurpuratus (Giraldes & Smyth, 2016). Previously, two lineages of lobsters, separated by the plume of the Amazon River, were identified by Sarver, Silberman & Walsh (1998). The lineage distributed south of this barrier and along the Brazilian coast was later validated as a distinct species, P. meripurpuratus, while the original term Panulirus argus was retained for the lineage from the Caribbean and North American coasts (Giraldes & Smyth, 2016). Therefore, all previous reports of P. argus in Brazil should actually refer to P. meripurpuratus.

Popular and trade names and their relationships with scientific species identification

The commercialization of shrimp in Bragança was characterized by significant variation in popular trade names, which lacked scientific-based criteria. As a result, some species were marketed under different commercial labels, as observed with P. isabelae and P. vannamei (“camarão cascudo” or “camarão piré”). This issue leads to imprecise estimates of the number of species commercialized in the region, as previously noted by other authors (Espírito-Santo & Isaac, 2012; Santana et al., 2020).

Only the samples of P. schmitti and P. meripurpuratus were identified by their specific popular names (“camarão branco” and “lagosta vermelha”, respectively) (Almeida et al., 2021; Carvalho et al., 2021). Other species were marketed under generic categories, such as the crabs U. cordatus (“caranguejo”) and C. bocourti (“siri”), even though these species are commonly known as “caranguejo-uçá” (Freitas et al., 2015) and “siri vermelho” (Jomar, 2021), respectively.

In some cases, species sold at the free fair were marketed under multiple denominations, as observed for P. monodon and M. sculptilis. For P. monodon, the accepted common name is “camarão tigre” (Benzie et al., 1995), but the term “camarão cascudo” was also used. In contrast, for M. sculptilis, the primary valid name is “camarão arco-íris” (Alam & Pálsson, 2021), although it was also recorded as “camarão tigre” in this study. The use of the name “camarão tigre” by local vendors for M. sculptilis may be attributed to the fact that it is a recently registered species in the region (Ferreira et al., 2023) and bears a strong resemblance to P. monodon.

Additionally, several species were not marketed under their primary common names in Brazil. For instance, P. isabelae was sold as “camarão rosa” (Franca et al., 2020), P. vannamei as “camarão cinza” (Freitas, Oliveira-Filho & Campagnoli, 2016), X. kroyeri as “camarão sete-barbas” (Franco & Santos, 2022) M. rosenbergii as “camarão gigante da Malásia” (Mohamed, Firuza & Subha, 2017), and P. laevicauda as “lagosta verde” (Lima & Andrade, 2017).

On the other hand, no popular denominations are available in the literature for M. equidens and the recently described shrimp species X. dincao. However, in Bragança, representatives of the Macrobrachium genus are commonly referred to as “camarão cascudo” (Espírito Santo et al., 2005), while species of the genus Xiphopenaeus are known as “piré” or “piticaia” (Santana et al., 2020), a naming convention also observed in the present study.

As demonstrated, there is a wide range of commercial names and a lack of standardization in their usage. Therefore, it is essential to establish guidelines to standardize the application of common names alongside their corresponding scientific names for commercial crustaceans. Regulating this nomenclature would enhance traceability and fishing statistics for various species at landing sites and in markets (Cawthorn, Baillie & Mariani, 2018; Cundy et al., 2023). Additionally, it would support conservation efforts, fair trade, and safeguard consumer rights by ensuring access to accurate product information (Delpiani et al., 2020; Pincinato et al., 2022).

Commercial and invasive carcinofauna in coastal Amazon: implications for fisheries management and conservation

Molecular identification has revealed that the diversity of commercial crustaceans has been underestimated due to the reliance on popular trade names. As a result, the current data have uncovered hidden levels of diversity within the regional carcinofauna, which were not previously documented (Freire, Silva & Souza, 2011; Santana et al., 2020). Had common names been used as identification markers, the two newly described species, P. isabelae and X. dincao, would likely have been misidentified as their sister species, P. subtilis and X. kroyeri, which have historically been reported in the coastal Amazon (Freire, Silva & Souza, 2011; Espírito-Santo & Isaac, 2012).

The reproduction and recruitment of certain commercial crustacean species are protected by fisheries closed seasons established by national regulations (Brasil. Lei no 11.959, de 29 de junho de, 2009). For lobsters such as P. meripurpuratus (formerly P. argus) and P. laevicauda, the closed season spans from November 1st to April 30th (regulation SAP/MAPA n. 221/2021), while the closed season for the crab U. cordatus runs from January to March (regulation SAP/MAPA No. 325/2020). Additionally, the fishing of penaeid shrimps (e.g., X. kroyeri and P. schmitti) is prohibited from December to February (regulation MDIC/MMA No. 15/2018). Despite these regulations, several crustaceans, particularly U. cordatus, are sold year-round at the free fair in Bragança, even during the closed season, reflecting illegal trade practices (Santana et al., 2020). The illicit exploitation of commercial species during their breeding season threatens the sustainability of fish stocks (Santana et al., 2020; Lima & Andrade, 2021). A similar situation has been observed in this region with teleosts and elasmobranchs (Martins et al., 2021; Santana et al., 2023), underscoring the high vulnerability of coastal Amazon fisheries due to unregulated overexploitation, which could undermine the long-term sustainability of fishing activities.

Regarding conservation status, none of the species recorded in this study are classified as threatened by the IUCN or national regulatory agencies (MMA n. 14/2022). Although there is evidence of population declines for U. cordatus, X. kroyeri, P. schmitti, P. argus (now validated as P. meripurpuratus), and P. laevicauda, reliable estimates of stock declines for these species are hindered by the lack of accurate statistical data (Pinheiro & Boos, 2016). Overexploitation, fishing during breeding seasons, and habitat degradation have been identified as the primary threats to wild populations of commercially important species, as well as to the long-term sustainability of fisheries resources (De Mitcheson & Erisman, 2011; Mitcheson et al., 2020).

Another critical issue revealed in this study is the substantial presence of alien species, which accounted for approximately 40% of the sampled crustaceans. Generally, these bioinvaders are carnivorous and pose significant threats to native species (Iketani et al., 2016; Ferreira et al., 2023). Such threats may arise through direct predation, intensified competition for environmental resources, and the potential transmission of pathogens to native fauna (Doherty et al., 2016; Svoboda et al., 2017). These direct and indirect pressures can negatively affect native crustacean stocks, ultimately undermining the sustainability of local fisheries and hindering conservation initiatives (Government of Brazil, 2000; Fuller et al., 2014).

Therefore, the negative impacts associated with the presence of alien species along the coastal Amazon are substantial, resulting in both environmental degradation and socioeconomic losses. Aquaculture activities and global maritime traffic are likely the main vectors of bioinvasion, highlighting the urgent need for effective monitoring and regulatory policies. In particular, measures aimed at controlling species introductions through ballast water discharge and aquaculture practices should be prioritized to prevent further dissemination of exotic species in the region (Loebmann, Mai & Lee, 2010).

Conclusions

In this study, the COI marker proved to be an effective tool for the discrimination and identification of crustacean species exploited by fisheries in Bragança. The results revealed a diverse commercial carcinofauna, including newly described taxa, invasive alien species, and indications of cryptic diversity within P. monodon.

Accordingly, molecular identification revealed previously underestimated levels of regional diversity among harvested crustaceans, which have traditionally been overlooked due to the widespread use of generic trade names. This issue is further compounded by the absence of national regulatory policies establishing standardized common names for commercially exploited crustacean species.

Therefore, the present data, together with future molecular inventories, should contribute to the establishment of a reference library for the carcinofauna of the Coastal Amazon, supporting the effective management and conservation of exploited species. Furthermore, the broad dissemination of scientific knowledge is essential to inform and promote public policies aimed at regulating local fisheries and trade, thereby ensuring accurate documentation of crustacean diversity and safeguarding regional biodiversity.

Supplemental Information

Supplemental Information 1 Maximum likelihood (ML) trees showing the haplotypes of crustaceans based in two regions (I and II) of the Cytochrome C Oxidase Subunit I (COI) gene, including the reference sequences from public databases

The statistical support values are shown on internodes.

Supplemental Information 2 The diversity of crustaceans from the free Fair (“Feira livre”) and collected in the wild

Supplemental Information 3 Genetic distance (K2P) among crustacean species using haplotypes from region I

Sequences from the NCBI database were included in the analysis.

Supplemental Information 4 Genetic distance (K2P) among crustacean species using haplotypes from region II

Sequences from the NCBI database were included in the analysis.

Supplemental Information 5 Banks of fasta sequences of portions I and II of COI from commercialized crustaceans

Additional Information and Declarations

Competing Interests

Author Contributions

Data Availability

The authors declare there are no competing interests.

Jefferson Sousa conceived and designed the experiments, performed the experiments, analyzed the data, prepared figures and/or tables, and approved the final draft.

Ítalo Lutz analyzed the data, prepared figures and/or tables, authored or reviewed drafts of the article, and approved the final draft.

Paula Santana analyzed the data, authored or reviewed drafts of the article, and approved the final draft.

Thais Martins analyzed the data, authored or reviewed drafts of the article, and approved the final draft.

Charles Ferreira analyzed the data, authored or reviewed drafts of the article, and approved the final draft.

Nicolly Santa Brígida performed the experiments, prepared figures and/or tables, and approved the final draft.

Josy Miranda analyzed the data, authored or reviewed drafts of the article, and approved the final draft.

Raimundo da Silva analyzed the data, prepared figures and/or tables, and approved the final draft.

Andressa J Barbosa analyzed the data, authored or reviewed drafts of the article, and approved the final draft.

Suane Matos analyzed the data, authored or reviewed drafts of the article, and approved the final draft.

Carla Mendes performed the experiments, prepared figures and/or tables, and approved the final draft.

Bruna Cardoso performed the experiments, prepared figures and/or tables, and approved the final draft.

Aline Silva performed the experiments, prepared figures and/or tables, and approved the final draft.

Ingrid da Silva performed the experiments, prepared figures and/or tables, and approved the final draft.

Jorge da Costa performed the experiments, prepared figures and/or tables, and approved the final draft.

Marcelo Vallinoto analyzed the data, authored or reviewed drafts of the article, and approved the final draft.

Iracilda Sampaio analyzed the data, authored or reviewed drafts of the article, and approved the final draft.

Grazielle Evangelista-Gomes conceived and designed the experiments, authored or reviewed drafts of the article, and approved the final draft.

The following information was supplied regarding data availability:

The sequences are available at NCBI: (region I) PP706442 to PP706467, (region II) PP708622 to PP708656.

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
