# Peer review of "Molecular identification based on mtDNA analysis of commercial crustaceans in the coastal Amazon: exotic species, cryptic diversity, and implications for sustainable fisheries in northern Brazil"

_PeerJ, doi:10.7717/peerj.19586_

## Round 0.1 · original submission · Major Revisions

Dear Authors, I kindly request you to revise the manuscript carefully according to each of the reviewers' comments. Re-reviewing this manuscript should enable the reviewers to approve its publication.

Reviewer 1 ·

Basic reporting

The researchers applied a DNA barcoding approach to identify crustacean species in one of the main fishing spots in Brazil -- Caeté River estuary. Based on molecular identification, they discovered 15 species from seven different genera and five families, as well as hidden cryptic diversity within Penaeus monodon. This work is important for our understanding of the true biodiversity and conservation management in the study area. However, I have comments on both methodological and theoretical parts of this manuscript, believe that it cannot be accepted in its current form, and suggest major revisions.

The English language should be improved; throughout the manuscript, there are some typos, unclear sentences, and untranslated texts. For instance:

1. line 57-58: it’s not clear how crab is different from crabs; please provide additional information for their separation; perhaps Ucides cordatus can also be called Mangrove ghost grab? The same applies to the subsequent mentions of the “crabs” and “crab” divisions, for instance at line 70 and line 198.
2. I suggest change term “portion I” and “portion II” to “region I” and region II” respectively, as the term “portion” might mean specific portions of proteins and non-coding sections, which control the gene expression.
3. Line 272: please translate the sentence to English.
4. Line 389: Differently to what? Perhaps, it’s better to use in contrast?
5. Line 413: please change “e” to “ and”.

This work includes sufficient background information on previous morphological studies of crustacean biodiversity in the studied area, however, it lacks some essential links and references on other cases of DNA barcoding initiatives on crustaceans in other areas, as well as references on already published cryptic diversity of P. monodon. Here are some comments:

6. From line 83: there are also known pitfalls of using the COI barcoding gene, such as maternal inheritance and particular proneness for introgression and therefore, usage of only this gene might lead to biased conclusions. It’s important to mention when discussing barcoding initiatives.
7. Did the authors compare the cryptic lineages of P. monodon to the already published cryptic lineages of the species? From table 1, it is seen that they are identical to the two cryptic species published by Yudhistira & Arisuryanti, 2019. In order to avoid misidentification, I would suggest referring these cryptic lineages in similar way it was described in Yudhistira & Arisuryanti, 2019 – as clade A and clade B – and mention that these cryptic lineages were reported previously.
8. Line 294: The same applies here – the supposed new species of P. monodon can be referred to the cryptic lineages described previously from Indonesia (or maybe somewhere else?). However, the novelty part can be in the fact that this cryptic species were not found in Brazil previously.
9. Line 331: it is indeed important to mention possible hybridization between native and alien species. However, this hybridization cannot be traced from only the mitochondrial COI gene, and therefore, I don’t see any reason for adding two paragraphs on hybridization here. The COI genetic similarity of the different species can be caused by hybridization (one of the pitfalls of the gene), but also by misidentification or other causes.
10. Line 369: again, this cryptic species was already reported previously.
11. In Discussion, I would also recommend providing comparisons with other barcoding initiatives for crustaceans in other geographical areas: what are the similar trends or differences that can be outlined from these comparisons?

The manuscript is structured according to the standard scientific requirements. Although the authors submitted the obtained sequences to the NCBI database, they are not available yet. Some comments on figures and tables, as well as article structure, include:

12. Fig. 1: please add the caption Penaeus monodon Lineage 2 to the respective branch
13. Fig. 4: please add popular common names along with Latin.
14. Please provide geographical coordinates for the two sampling localities used in this study (either in Table or in the text).
15. Also, please specify how many individuals from each species were sequenced (either in the Table or in the text).
16. Line 205: the next two paragraphs can go to methods.

Experimental design

While the research questions are well-defined, the methodological part requires some revisions. In order to improve this work, I provide some comments as follows:

17. line 120: The authors mention that tissues were taken from each specimen and stored in 70% ethanol. How fresh were these samples? On line 216, the authors wrote that some samples were cooked or salted. Perhaps it’s better to mention it in methods.
18. From line 156: Did authors align sequences of different portions separately? Were there any overlaps between the sequences from different portions? If align sequences from different portions together, what would be the length of the trimmed alignment? This information is important to understand why following separate phylogenetic analyses were performed.
19. Why did authors use NJ trees with the K2P model, while Bayesian phylogeny was built using the GTR+I model? On line 183, the authors specify that the best-fitted model for the datasets was GTR+I, so this model should be used in all phylogenetic inferences. I would suggest building ML trees under best-fitted GTR +I model, since ML uses more complex evolution models and is known to be stronger and more reliable than NJ reconstructions. The same applies to the genetic distances, which can be calculated as simple p-distances or model-corrected using the appropriate model.
20. Please add the table of intra- and interspecific genetic distances to the supplementary material and refer to it in the text. This is important when discussing species delimitation and diversity.
21. In addition, please also mention why the trees were middle-rooted.
22. When studying the native and alien species diversity in the commercialization and conservation context, it is important to discuss intraspecies diversity stats. I would suggest adding values of intraspecies nucleotide and haplotype diversities to determine genetic variation. This can be an important outcome of this work, since there are hints of the high and low diversities from Table 1, where most of the haplotypes are derived only from single sequences, while some other haplotypes include up to 19 sequences.
23. Since the authors are dealing with molecular species identification and delimitation, I would also suggest adding species delimitation methods such as ABGD, ASAP, and GMYC to justify the presence of the 15 species and hidden cryptic diversity.
24. Line 305: it’s not clear how the concatenated dataset would look like. So, there are no overlaps between portions? This information should go to the methods.

Validity of the findings

Overall, I believe that this work could bring new knowledge on the crustacean biodiversity in the study area, however, it requires major changes and rewrites in both methodological and theoretical parts. My main concerns include:

25. links to the previous works on P. monodon cryptic diversity. The authors state that they discovered new sympatric cryptic species of P. monodon, however, the obtained sequences are identical to the sequences of the cryptic species reported from Indonesia by Yudhistira & Arisuryanti, 2019 and therefore these species should be refereed more accurately.
26. Could authors also discuss why the position of Penaeus isabelae differed in the separate trees? I wonder if the result will be the same when using ML reconstruction.
27. The authors do not mention the intraspecific diversity. However, it is one of the key parameters for studies of the biodiversity of native and alien species in the context of commercialization and conservation.

Reviewer 2 ·

Basic reporting

Overall, I think this article has potential, but the authors did not effectively convey their ideas. The analyses need to be redone and the sentence structure requires significant improvement. Some of the suggestions for improving the writing have been incorporated into the revised manuscript.

Experimental design

The use of the term 'haplotype' is confusing. What is the paper intended to achieve? Is it focused on identifying haplotypes in the context of population genetics, or is it aimed at exploring species diversity? If the latter, other suitable analyses should be included, such as running an ML tree and using species delimitation software like Assemble Species by Automatic Partitioning (ASAP), bPTP, or GMYC. The experimental design is not clear.

Validity of the findings

The findings are valid; however, the way they are written and conveyed is not clear. I understand what the authors intended to do, but they need to reanalyze their data and revise their writing.

Additional comments

All the comments for improving the manuscript have been incorporated into the revised version.

Annotated reviews are not available for download in order to protect the identity of reviewers who chose to remain anonymous.

Reviewer 3 ·

Basic reporting

Articles must improve in the abstract, materials and methods, and results sections. Need to be written in details in the materials and methods sections and the results part should elaborate on COI and cluster analysis

Experimental design

Experimental design is not clear. Provide primer details for two COI regions and protocol

Validity of the findings

I am not able to access the NCBI database based on the accession number provided in the materials and methods section

Additional comments

Abstract: In abstract section, not clear the methodology part
Materials and Methods: There is no two COI primer details in methodology section; I am not able to access the NCBI database based on the accession number provided in the materials and methods section
Results: There is no clarity in the results section. Need to write in elaborately about genome size and cluster analysis

---

## Round 0.2 · Major Revisions

Dear authors, The article needs to be seriously revised because the reviewer feels that the methodological part of this article needs to be carefully revised; there are many concerns regarding the appropriate methods and interpretation of the results. I kindly ask you to revise the manuscript very carefully according to each of the reviewer's comments.

Reviewer 1 ·

Basic reporting

The quality of the article significantly improved compared to the previous version, however, I still have some comments and concerns about the analysis (see in experimental design). As to the basic reporting, the manuscript is well-structured and includes all necessary parts. Here are some comments regarding the text of the article:

- the authors say that the've obtained 16 commercial denominations and 15 species. Does it mean that some species have two commercial denominations? This is not clear.

-line 35: The authors use fragments and then sequences, it is better to use one term (sequences) and rewrite the sentence.

-line 240: the common name for lobster species appeared here but was not mentioned in the introduction starting from the line 62.

Table 1: the last column referred to the ncbi sequences, to which the obtained sequences were compared? It’s not clear now, so I would suggest changing the column name.

Line 389: region 1 or region I?
Line 390: region 2 or region II?

-As a general comment, I would advise you to read and check the manuscript text with a native speaker. Also, please check the reference list carefully.

Experimental design

My main concerns are about the tree reconstruction methods and the models.

- Why did the authors keep the NJ tree? Compared to the ML and BI reconstruction methods, NJ trees are less trustworthy, so I would suggest using only ML and BI trees; no need for NJ.

- The authors added ML trees to the analysis, yet there is no mention of ML trees in the results.

- The authors say that the NJ tree is used to represent the results (line 280), but in Figure 2 and 3 captions, it's mentioned that they are BI trees. Which trees were used where? This is confusing.

- Why the trees were middle-rooted? Is it possible to use an outgroup for the analysis?

- Why models for BI and ML trees are different? Were models estimated for the different datasets from two COI regions? Why are distances calculated using different K2P model? If one picks a model for the dataset, it’s better to perform all analyses using this model.

- Is there any difference between samples from natural environments and the market? Could the quality of the material affect the quality of sequences?

Validity of the findings

The findings could be valid if the analysis is redone and communicated more clearly.

Reviewer 3 ·

Basic reporting

No Comment

Experimental design

No Comment

Validity of the findings

No Comment

Additional comments

No Comment

---

## Round 0.3 · Major Revisions

Dear authors, I kindly ask you to make changes to the manuscript in accordance with the reviewer's comments.

Reviewer 1 ·

Basic reporting

Dear authors,

Thank you for the changes implemented in this version of the manuscript. However, I still advise to proofread the text once again with a native speaker and check the text for typos/grammar/style.

Line 34: rewrite this sentence to "We obtained a dataset comprising 16 commercial denominations and 151 sequences, including 38 – from region I (barcode region) and 113 – from region II of the COI gene.”

Line 194: sentence is not clear, please rewrite.

Line 833, 924, 926: Please check the references and correct them according to the guidelines.

Experimental design

Line 209: In the IQTREE software, one can specify the model, so I suggest that authors use the same GTR+I model for both ML and BI analyses.

Line 212: Same for the distances. If the best model for the datasets is GTR+I, I would recommend calculating the distances using this model too. In case there is no such model in Mega, it is possible to do it in the PAML software, for instance.

Line 212: Did the authors calculate net distances between the clades/species or total distances?

Please add ML trees to the supplementary material.

Validity of the findings

--

Additional comments

--

---

## Round 0.4 · accepted · Accept

Dear Dr. Evangelista-Gomes, I am pleased to inform you that your article has been accepted for publication in our journal. I look forward to continuing your interesting and practically important work on the study of crustaceans in South America.

Reviewer 1 ·

Basic reporting

Dear authors,

thank you for the updated version. I agree with all the changes made.

Experimental design

--

Validity of the findings

--

Additional comments

--